# Radiation-Induced Defects and Effects in Germanate and Tellurite Glasses

**DOI:** 10.3390/ma13173846

**Published:** 2020-08-31

**Authors:** Mikko Hongisto, Alexander Veber, Yannick Petit, Thierry Cardinal, Sylvain Danto, Veronique Jubera, Laeticia Petit

**Affiliations:** 1Photonics Laboratory, Tampere University, Korkeakoulunkatu 3, 33720 Tampere, Finland; alexander.veber@tuni.fi (A.V.); laeticia.petit@tuni.fi (L.P.); 2CNRS, University of Bordeaux, Bordeaux INP, ICMCB, UMR 5026, F-33600 Pessac, France; yannick.petit@icmcb.cnrs.fr (Y.P.); Thierry.Cardinal@icmcb.cnrs.fr (T.C.); sylvain.danto@icmcb.cnrs.fr (S.D.); veronique.jubera@icmcb.cnrs.fr (V.J.)

**Keywords:** germanate glass, tellurite glass, radiation treatment, defects, structuring

## Abstract

This review focuses on the radiation-induced changes in germanate and tellurite glasses. These glasses have been of great interest due to their remarkable potential for photonics, in terms of extended transmission window in the mid-infrared, ability of rare-earth loading suitable with a laser, and amplification in the near- and mid-infrared or high nonlinear optical properties. Here, we summarize information about possible radiation-induced defects, mechanisms of their formation, and the influence of the glass composition on this process. Special attention is paid to laser-induced structural modification of these glasses, including possible mechanisms of the laser-glass interaction, laser-induced crystallization, and waveguide writing. It is shown that these methods can be used for photostructuring of the glass and have great potential for practical applications.

## 1. Introduction

Glasses play a key role in many areas of modern life. The fundamental understanding of the photoresponse of the glass to radiation has allowed the development of novel glasses with tailored photoresponse as, for example, radiation-hardened optical fibers based on glassy silica [1]. High-energy radiation can induce multiple physical and chemical modifications in materials. It includes among other modifications in crystallinity [2,3] and bond structure [4], lattice defects [5], optical properties [5], spectroscopic properties [6], electrical properties [2], and surface and interface morphology. These modifications are a consequence of lattice defect creation, migration, and recombination, the kind and extent of defects depending mostly on the substrate material, radiation type, dose, and energy fluence. Although the photoresponse has been widely investigated for silica [7,8], chalcogenide [4,9], and phosphate glasses [10,11], fewer studies have been reported on the radiation treatment of heavy metal oxide (HMO) based glasses. These are often described as glasses containing TeO_2_, Sb_2_O_3_, GeO_2_, Ga_2_O_3_, and/or Bi_2_O_3_ glass formers, to cite a few.

The objective of this review is to provide details on the defect formation mechanism and the radiation-induced variation of properties in tellurite and germanate glasses, such as their optical and structural properties. Besides creating point defects and modification of the glass network, radiation treatment can induce crystallization or bubble formation [10]. In the selected papers, the glasses were irradiated using different sources ranging from the ionizing (alpha, beta, gamma, and X-rays) to the near-infrared ultra-short pulse laser radiations [12]. With laser radiation treatment, the energy is controlled and focused, and thus has been used extensively to write structures, such as active [13] and passive [14] waveguides in glasses, for example. Another objective is to propose strategy to engineer radiation-tolerant HMO glass photonic devices and/or to mitigate their damages (ex: annealing) [11].

This review is organized as follows: at first (Section 2), a general introduction to HMO glasses and to their physical, chemical, optical, and structural properties is presented. After (Section 3), the mechanisms of energy deposition, structural relaxation, and the different defect generation processes are introduced. Finally (Section 4), the use of radiation treatment to locally structure the glasses is described.

## 2. Description of the Tellurite and Germanate Glasses

Heavy metal oxide (HMO) based glasses have been popular choices for applications in the telecoms and mid IR bands (up to around 5 µm), since the most common silicate glasses cannot be used in this spectral region [15]. Since the discovery of several HMO glasses, the tellurite and germanate glasses have demonstrated growing interest, as these glasses possess wide transmission region, good corrosion resistance, low phonon energy, and high linear and nonlinear refractive index [16]. As opposed to silica glass, these glasses are transparent beyond 2 µm, where silica glass become opaque. They are also good hosts for rare-earth (RE) ions as they can also incorporate a large amount of rare-earth ions without clustering [17,18,19]. These HMO glasses are especially good hosts for mid-infrared emitting optical centers, e.g., Er^3+^ emitting at 2.7 µm, having potential applications in surgery, remote atmospheric sensing, and eye-safe laser radar because of their spectral overlap with regions associated with strong water absorption [20].

Tellurite and germanate HMO glasses have been identified as good glass hosts also for metal or semiconductor nanoparticles (NPs) with the objective to enhance their optical properties provided by the surface plasmon resonance (SPR) mechanism [21,22]. Applications have been found for these nanoparticle/glass structures in photothermal therapy [23], medical sensing of antibodies [24], and Raman spectroscopy [25], just to cite several examples. Metallic NPs containing HMO glasses are also promising materials for use in nonlinear optical devices, as the NPs can also be used to enhance the linear and nonlinear properties of the glasses [26,27]. The introduction of silver has appeared as the most efficient and fruitful route toward the functionalization of glasses with distinctive properties. Since the first reported study on the nucleation of silver in germanate (PbO-GeO_2_) and in tellurite glasses in [28] and in [29], respectively, precipitation of Ag nanoparticles in germanate and tellurite glasses has been of great interest. For example, considering tellurite glasses [30] have synthesized the Er^3+^/Yb^3+^ codoped tellurite glasses with silver NPs through melt-quenching method, and a large enhancement in visible upconversion emissions (539, 557, and 675 nm), the near-infrared emission (1.5 µm) along with the mid-infrared emission (2.7 µm) was found with the precipitation of silver NPs. In [31], it has been reported the surface-enhanced Raman scattering and plasmon-enhanced emission by silver NPs in Er^3+^-doped zinc-tellurite glasses. An enhancement by about 10 times in Raman signal and up to three times in upconversion emissions (520, 550, and 650 nm) of Er^3+^ was demonstrated. In germano-bismuthate glasses, a significant enhancement of the Tm^3+^ fluorescence intensity at 1.47 µm in Ag_0_ NPs embedded glasses could be obtained [32]. Similar enhancement was also obtained in Er^3+^/Yb^3+^ doped glasses [33]. All the aforementioned fluorescence enhancement of doped rare-earth ions is mainly attributed to the local field enhanced effect of metallic NPs. The deep understanding of the formation mechanisms of silver nanoparticles in host matrix is essential to achieve new silver species with proper size and shape, inter-particle distances, and volume concentration. Gold insertion was also considered as a luminescence sensitizer of rare-earth ions, such as in the Er^3+^-doped TeO_2_-ZnO system by [34]. These metallic nanoparticles were found to enhance the photoluminescence due to presence of the RE ions in the proximity of an array of silver nanoparticles.

### 2.1. Elaboration Methods of HMO Glasses

HMO glasses are generally obtained by a standard melt-casting method under different atmospheric conditions, the most commonly employed crucibles being platinum (Pt), gold (Au), or corundum (Al_2_O_3_). Both the atmosphere and the crucible depend on the glass composition to be melted. For example, the tellurite glasses tend to dissolve Pt from crucible as Pt is not inert when in contact with TeO_2_ [35]. The glass batches can be melted in different atmospheres. An atmosphere of pure oxygen at ambient pressure would avoid reduction of TeO_2_ during the melting [36]. A review on the preparation of tellurite glasses was recently published and can be found in [37]. Sol–gel technique has been also used for the preparation of HMO glasses. The main advantages of this technique compared to the melting–quenching method are (1) the low processing temperature, (2) the control of the purity and homogeneity of the glass, and (3) the flexibility of the glass composition. Recently, aerodynamic levitation technique (ADL) has been reported to be a suitable technique to prepare HMO glasses [38]. In ADL technique, the melt is levitated by a gas flow without the use of crucible before being solidified. This technique is often used to prepare glasses with low glass-forming ability to vitrify in bulk form, especially when preparing glasses with large amount of RE ions.

### 2.2. Glass Network Structure of HMO Glasses

It is well known that the various properties of the glasses are dictated by their structure. Therefore, it is crucial to understand how the atoms are bonded to form the network. Tellurite glasses have been under investigation since their discovery by Stanworth in 1952 [39]. The main glass network is constituted of tellurium ad oxygen atoms, belonging to the group VIA in the periodic table of elements [16]. The network can be formed by TeO_4_ (trigonal bipyramid), TeO_3_ (trigonal pyramid), and the intermediate TeO_3+1_ polyhedron. Each unit carries a lone pair electron (LPE) [40], which is thought to contribute to the large linear and nonlinear refractive index of the tellurite glasses. As shown in Figure 1, the four oxygens in the TeO_4_ units are coordinated to one tellurium atom to form a trigonal bipyramid (tbp).

The two equatorial and two apical oxygen sites are bridging oxygens (BO), while the third equatorial site is occupied by the LPEs in the valence band of tellurium. In the TeO_3_ structure, two oxygens are bridging oxygen and one oxygen is one non-bridging oxygen (NBO) double bonded with a tellurium atom (Te=O). During the formation of the glass network, the site for an oxide ion and an LPE can interchange mutually with four bridging oxygen sites leading to bond deformation and packing of structural units, which is a unique characteristic of tellurite glasses. Others have described the structure in terms of Q_m_^n^ units as represented in Figure 2. N is the number of bridging oxygen bonded to the central atom and m is the coordination number, as of oxygen around tellurium, which is three or four [42]. The addition of modifier cation was reported to lead to the transformation of the Q_4_^4^ to Q_4_^3^ (TeO_4_ to TeO_3+1_) [43].

The glass formation and structural analysis of tellurite glasses have been intensively studied [44,45]. A review on the structural properties of tellurite glasses with different compositions can be found in [41].

The network of pure germanate glass is formed by GeO_4_ units which share their corners, the Ge atom being covalently bonded to four bridging oxygens. The structural units, along with tetrahedra, can be trigonal bipyramids and octahedra, where the central atom is germanium, surrounded by five or six oxygen atoms, respectively, as depicted in Figure 3.

As opposed to silica glass, the physical properties of germanate glass exhibit extrema depending on the alkali oxide content. This phenomenon is known as the “germanate anomaly effect” which was first reported in 1962 [47]. The first model to explain this anomaly effect was proposed in 1964 [48]. Murthy et al. suggested that the coordination of Ge gradually changes from GeO_4_ to GeO_6_ and then from GeO_6_ to GeO_4_ with the progressive introduction of alkali oxide. This change in the Ge coordination was confirmed using X-ray, neutron scattering [49,50], and also using extended X-ray absorption fine structure (EXAFS) [51,52]. However, the presence of GeO_5_ or GeO_6_ in the germanate network is still uncertain as these characterization tools cannot distinguish between fivefold and sixfold coordination of Ge. According to molecular dynamics study [53] and neutron diffraction with high real-space resolution [54], the higher Ge coordination is suspected to be GeO_5_. However, others reported that non-bridging oxygen (NBO) atoms would occur when the alkali oxide content is below 10 mol%. The germanate anomaly phenomenon would then be related to the transformation of different member rings [55,56]. In this model, the addition of alkali oxide is suspected to lead to the formation of a three-membered ring which converts to a four-membered ring generating non-bridging oxygen with further addition of alkali oxide. Recently, a new insight into the mechanism of germanate anomaly was reported in [57]. Three Ge structural units (fourfold, fivefold, and sixfold) were found in the K_2_O-GeO_2_ glass system with a K_2_O content between 11 and 20 mol%, whereas the network contains only two Ge species (fourfold and sixfold) when the K_2_O content is larger than 20 mol%. The main structural modification process for the content of K_2_O > 0.25 is the depolymerization of the germanate network through NBO formation in various Q^n^ species. For more information about germanate anomaly, the reader is referred to [58]. Additional element as RE added in the glass network could also influence the local structure of germanate glasses. If variation of RE concentration does not modify the molecular vibrational part of the Raman response, which indicates that no specific new site is stabilized, these doping elements impact the non-bridging anions in their vicinity. This results in a variation of the boson intensity peak which has been described in [59]. Tuning of the glass transition and crystallization temperature could also be managed by controlling the RE concentration [3].

## 3. Defects Formation in Germanate and Tellurite Glass Due to Radiation Treatment

It has been known that ionizing radiation affects the glass network by creating defects, which in turn affect the properties of glasses with changes in the optical properties receiving the most interest [5]. These defects can, for example, be used for radiation measurement in sensing, cause absorption losses in fibers [60], or act as luminescent centers and replace phosphorescent compounds in lighting applications [61]. To better understand the glasses’ performance in these applications, it is therefore imperative to understand the defects and how they are formed.

### 3.1. Types of Defects and Their Characterization

The types of defects that have been discovered in tellurite glasses are shown in Figure 4.

The Te-NOBHC is a central Te atom linked to three bridging oxygens and one oxygen being a non-bridging one with positive charge indicating a trapped hole. The TeEC and V_0_ defects have been attributed to an electron and a hole trapped in an oxygen vacancy, respectively, while the Te is connected to three oxygens [62]. A schematic of different defects formation is shown in Figure 5. A modifier-related trap is observed when an electron is trapped near a modifier atom. The TeOHC is deemed to be an intrinsic defect consisting of a hole trapped in the bridging oxygen [63].

While the effects of various types of radiation on tellurite glasses have been studied in detail [64,65], the characterization of the formed defects have received significantly less attention. The existing defect-related studies are sparse and focused mainly on the EPR measurements [62,63,66]. No direct attributions of optical absorption bands to specific defects have yet been made. However, there exists research results describing absorption caused by radiation-induced defects [64]. Table 1 shows the defects and their respective electron paramagnetic resonance (EPR) g-values.

Various defects have been observed in germanate glasses. The defects can be classified as intrinsic or extrinsic. Intrinsic defects are formed during the preparation of the glass, whereas the extrinsic defects are due to ionizing radiation breaking bonds or changing the valency of modifier ions. The defects are further classified based on their structure, whether the defect is formed with a positive hole, i.e., absence of electron, or a trapped excess electron. Formation and photoresponse depends on the structure and type of the defect [67]. Upon prolonged exposure to irradiation, more and more extrinsic defects are formed, and the number of intrinsic defects decreases. This causes a reduction in the defect-induced absorption band in a process known as bleaching [68]. The structures of germanium-related defects usually encountered are presented in Figure 6.

Intrinsic germanate defects are expected to form at the glass preparation stage [67]. These defects are classified as neutral oxygen monovacancy and divacancy color centers, NOMV and NODV, respectively (see Figure 6). These linked Ge atoms are precursors to further extrinsic defects caused by irradiation [69].

The germanium related lone-pair center (GLPC) is comprised of a germanium atom linked to two oxygens and a lone pair of electrons. The entire system has a positive charge, resulting in a (GLPC)^+^ defect. Germanium electron trapped center (GEC) is a fully coordinated Ge atom with an extra electron trapped at the center. The Ge E′ (E′ (Ge) in some texts) is a neutral color center defect where a Ge is linked to three oxygens and the fourth link is replaced by a lone pair of electrons. The germanium-related non-bridging oxygen hole center (Ge-NBOHC) consists of a central Ge atom linked to four oxygens. One of the oxygens is an NBO atom and has a positive charge due to a missing electron. The GeO_3_^+^ defect is a counterpart to a Ge E′ defect, a Ge linked to three oxygens, and a residual positive charge. The structures of these defects and the proposed mechanisms of their formation are presented in Figure 7.

It is proposed that the extrinsic defects are formed via two different processes: one- and two-photon reactions. In the one-photon process, in Figure 7a, a UV photon (243 nm/5.1 eV) forms an electron and pair of Ge E′ and GeO_3_^+^ defects. This is the result of a NOMV Ge-Ge bond breaking. NODVs do not break under UV irradiation; rather, the center relaxes from the excited state through photoluminescence. In the two-photon reaction, it is proposed that the NOMV is changed into two Ge E′ defects as shown in Figure 5b. NODV and normal GeO_2_ units can also react and form a mix of NOMV, GLPC, and GEC defects through an intermediate state as shown in Figure 5c [69]. However, it should be noted that an alternate model for the two-photon reaction has been proposed by [70]. This model does not contain NODVs, present in Figure 7c, but rather, it assumes that GeO_2_ and neutral GLPC react to form a pair of GEC and (GLPC)^+^ defects.

The various defects are usually referred to as color centers, meaning they absorb light in the UV-visible range. Although some defects are optically inactive, most have bands in the UV with some in the visible range. Therefore, a UV-Vis absorption measurement offers an easy and effective way to study various optically active defects [8]. Another common technique used to complement the absorption measurements is the electron paramagnetic resonance spectroscopy (EPR). It is especially useful in cases where the defect is optically inactive or the absorption band of one type of defect overlaps with another defect’s band. Other techniques such as photo- and thermoluminescence, PL and TL respectively, have been reported in some papers [71,72] and can be used together with absorption and EPR measurements. Table 2 shows the EPR g-values and corresponding absorption peak parameters for different defects.

### 3.2. Impact of Glass Composition on Defects Formation

The sensitivity of HMO glasses and the type of defects induced due to irradiation of the material, depends on the chemical composition. It is clear for instance that formation of Te-related defect is impossible in Te-free glass. At the same time, different types of defects can be induced in presence of several network formers.

Irradiation of pure GeO**_2_** glass results in formation of Ge E′ and oxygen excess centers: Ge-NBOHC and Ge-O**_3_^+^** [75]. Gradual introduction of GeO**_2_** in other glass systems, e.g., in gallate glass, also results in formation of Ge-related defects upon the irradiation, as it was demonstrated for 3CaO-2Ga**_2_**O**_3_**-xGeO**_2_** glasses (x = 0, 3, 4) upon γ, X-ray, and UV irradiation [76]. At the same time, the number of the induced centers depends on the pre-existing units in the glass network. A detailed investigation on number of Ge-E′ and Ge-NBOHC induced under irradiation was done for xNa_2_O-(1-x)GeO_2_, where x = 0–0.35 glasses [77]. The EPR center density was investigated as a function of the modifier content (Figure 8). It can be seen that the number of both defects increases coherently with the modifier content up to ~5 mol% (region I); that at higher concentrations up to 20 mol.%, the concentration of Ge-NBOHC continues to increase while amount of Ge-E′ decreases (region II); and that finally, a further increase of Na**_2_**O results in saturation or even slow decrease in Ge-NBOHC centers (region III).

Apparently, at low modifier levels, only slight changes in the glass network occur, and the following mechanism is responsible for the formation of the defect pairs:(1)≡Ge−O−Ge≡+hν →   ≡Ge−O⬚●+G⬚●e≡

In their original work, Azzoni et al. explained the higher yield of the reaction with increasing Na**_2_**O content in region I by network densification, without detailed investigation of its microscopic origin [77]. One may note that this reaction is identical to the one proposed for silicate glasses, where it was found that the number of defects induced is proportional to number of strained bonds in silica network, which are associated in particular with presence of threefold rings [78]. Investigation of Na**_2_**O-GeO**_2_** glasses by means of Raman spectroscopy revealed that number of three-membered GeO**_4_** rings increases up to ~10 mol% of the modifier content [79], allowing to assume that the increasing amount of defects in region I of both the types associated with low-membered rings occurs in the same manner, as it happens in silicate glasses.

Further increase in Na_2_O content leads to formation of significant amount of NBOs in the glass network and the following reaction is prevalent in regions II and III:(2)≡Ge−O−Na++hν →   ≡Ge−O⬚●+N⬚●a

Maximal yield of the Ge-NBOHC is observed in the range 15–25 mol% of Na_2_O, which matches with the region of the maximal strain in the glass network [79]. Despite an even larger amount of NBOs at Na_2_O concentration beyond 25 mol%, the defects formation yield does not change significantly, which can be explained by relief of the bonds strain [79].

The higher yield of defects upon irradiation in the glasses with larger amount of modifier seems to also be valid for Te-glasses [62] as well as for Si-glasses [78]. Thus, the amount of preexisting NBOs in the glass network before the irradiation could be one of the key factors influencing the overall defect yield and should be also valid for other glass systems. However, many HMO glasses still need to be investigated experimentally. Another important factor is strain of the bonds in the glass network, which also can be assessed by modification of the glass chemistry and often can be related to glass density.

However, it is possible to partially control the generation of color centers and improve radiation hardness of HMO glasses without significant changes in its structure. This can be achieved by low level doping with an additional element. It was found that introduction of Tb, Pr, and Ce can significantly decrease amount of the defects induced upon irradiation of GeO_2_-Gd2O_3_-BaO glasses (Figure 9), while low levels of the dopants should not significantly change the glass network [80].

The addition of cerium is known to increase the radiation hardness of different glasses, including germanate [80,81] and tellurite glasses [82]. Usually, two forms of cerium are stabilized simultaneously in glass, namely Ce^3+^ and Ce^4+^ ions. It has been found that Ce^3+^ and Ce^4+^ are capable of trapping electrons and holes induced upon irradiation of glass as follows [83,84]:$$Ce^{3+}+hole^{+}\to Ce^{3+\;+}\;Ce^{4+}+e^{-}\to \left[Ce^{4+}\cdot e^{-}\right]$$

The irradiation results in the formation of Ce-related defects, rather than electron or holes centers associated with the glass network, such as Ge-EC and Ge-NBOHC in the case of germanate glasses. The Ce-related defect centers do not have absorption bands in visible and NIR ranges of optical spectrum, consequently addition of Ce helps to minimize photodarkening effects in glass. Moreover, it is shown that newly formed Ce^3++^ and [Ce^4+^∙e^−^] centers can quickly transform back to original Ce-states, even at ambient conditions [84,85]. Thus, Ce-doping results in the permanent enhancement of glass radiation hardness. This is used in commercial fiber lasers to protect against defects formed by the laser itself. The defect would absorb pump light and therefore reduce laser power over time [85,86].

An increase of the glass radiation hardness was also observed on codoping of HMO glasses with other polyvalent elements such as Tb, Pr, Ni, and Sn [80,81,82,87]. Significant influence of rare-earth elements (REE) on the defect formation mechanism in HMO glasses can be also detected by variation of their thermoluminescence (TL) properties. It was shown that the doping of tellurite glasses by REE can strongly modify their TL induced under γ irradiation [22]. The dependence of glass properties on used REE dopant can be explained by difference in the efficiency of the defect trapping and their release during heating afterwards among the elements [22,88].

However, in contrast to Ce, these elements often require higher doping concentrations; possess optical bands in Vis-NIR optical ranges; and have different effects, depending on the matrix glass composition or glass redox, which makes Ce the most common dopant used to improve resistance of glass against irradiation. In particular, it has been found that Ce codoping can be an effective way to reduce photodarkening in HMO glasses—this approach has been already widely used for prevention of photodarkening in active Yb/Er doped silicate glasses [21,89].

Changes induced in glasses under irradiation are often reversible and the initial glass state can be restored by annealing of the irradiated material at elevated temperatures [16]. At sufficiently high temperatures, the defects can overcome the trapping energy barrier and escape the trap, which is increasing mobility of the defects and their recombination rate. Partial recombination of the defects and recovery of the glasses can happen even at room temperature. In particular, it was shown that TL-response of γ-irradiated REE-doped Te-glasses decreases for up to 15% in two months [22], i.e., a considerable decrease of the trapped defects with time is observed. Moreover, fading of the TL with time was found to depend on the REE dopant and Dy-doped glass being the most stable from this point of view [22]. This implies that use of different dopants can allow controlling stability of the radiation-induced defects in glasses and tailor properties of the material for a specific application—for instance, glasses with long-term defect stability can be used for dosimetry purposes [25].

In addition to temperature, visible light has been used to reverse the defect formation. In a germanosilicate fiber preform containing intrinsic defects, 3.5 eV/330 nm UV radiation was used to bleach luminescence caused by the defects. Here, the absorbing center, presumably related to the Ge-NBOHC, absorbs the bleaching photon and transfers the energy internally to destroy a defect with much higher absorption band [90].

## 4. Local Changes in the Glasses Due to Radiation Treatment Using Short Pulsed Lasers

In the following section, the use of ultra-short pulses radiation treatment to locally structure the glasses is described. In recent decades, ultra-short pulsed lasers have been of great interest due to their unique advantages in three-dimensional processing of materials. Femtosecond laser-induced photochemistry has been intensively investigated in a large variety of glass matrices, including silicates [91], aluminosilicates [91], aluminoborates [92], chalcogenides [93], and silver-activated phosphates [12]. Indeed, focused femtosecond laser pulses with energies of a few hundreds of nanojoules have become a key tool to modify the physical properties of glass in three dimensions (3D). Because of the nonlinear behavior of the interaction, the energy deposited by a focused femtosecond pulse is confined inside the focal volume. The high repetition rate femtosecond laser irradiation promotes not only the generation of defects or photodarkening of glass but also microstructure rearrangement associated with density changes, ion migration, or phase transition [92].

### 4.1. Laser-Induced Structural Modifications

Structural modifications are often observed upon high-intensity ultra-short pulsed laser irradiation of HMO glasses. The effect of the laser irradiation depends on the glass composition as well as on the laser beam parameters used during the irradiation, such as average energy, pulse duration, pulse repetition frequency, and beam focusing.

In tellurite glasses, fs-laser irradiation can cause transformation of trigonal bipyramids to trigonal pyramids Te-units. The latter are characterized by the shortening of Te-O bond lengths and result in the local increase of glass density and refractive index [94]. Migration of elements is another effect observed in tellurite glasses upon laser irradiation [94,95]. For instance, redistribution of Te and Na was observed in 50TeO_2_-20P_2_O_5_-20Na_2_O-5ZnO-5ZnF_2_ glass (Figure 10). The migration direction of elements is found to depend on the laser fluence [94] and laser intensity distribution [96], which shows that ion diffusion mechanisms are rather complex and still require further investigation.

In germanate glasses, it was found that single 150 fs pulses with energies up to 600 nJ (λ = 800 nm) do not change Ge-coordination but cause an increase in the number of three-membered rings of (GeO)_3_. By comparing a permanently densified Ge-glass (see Figure 11) and amorphous GeO_2_ with different thermal history, it was established that the densification mechanism is prevalent under this irradiation regime and the degree of this modification is proportional to the laser power [97]. At the same time, if the pulses are too close to each other in space, the glass crystallizes in a hexagonal GeO_2_ form and no presence of the high-pressure tetragonal GeO_2_ can be detected. This highlights the high local temperature accumulation in the sample, which causes devitrification of the material. This effect was explained by a local decrease of the thermal diffusivity and hence accumulation of the heat in the laser modified area [98]. Similar changes were observed for 15Na_2_O-85GeO_2_ (mol%) glass [99], which allows to suggest that these mechanisms also remain dominant in modified Ge-glasses. Nevertheless, higher fluence laser irradiation can also cause Ge and O ions separation with the formation of the molecular O_2_ species inside the glass [100] or the migration and redistribution of the elements in the glass network [101,102,103].

### 4.2. Photostructuring

Sophisticated localized structures can be written in glasses using high-repetition rate ultrafast lasers in a controlled manner.

For example, laser irradiation can be used to produce waveguides suitable for amplifiers, couplers, splitters, and sensors. Depending on the laser and the glass, the radiation treatment can lead to a refractive index change, a birefringent refractive index modification, or voids due to micro-explosions. The first waveguide amplifier in tellurite glass was reported in 2014 [94]. The increase in the refractive index was related to the migration of tellurium towards the irradiated region. The sodium migrates to the tellurium deficient zone and forms a relatively low index change region. Refractive index dots with lower refractive index were also generated in tellurite glasses in the TeO_2_-Na_2_O-Al_2_O_3_, TeO_2_-Na_2_O-GeO_2_ and TeO_2_-Na_2_O-TiO glass systems due to the direct heating of the glass with the laser beam spot [104]. The pattern of the refractive index dots was almost equivalent to the beam size.

D.M. Da Silva et al. reported on the inscription of single-line active waveguides in Er^3+^-doped germanate glass in the GeO_2_-PbO-Ga_2_O_3_ system using a femtosecond laser delivering pulses of 80 fs duration at 1 kHz repetition rate [13,105]. In this case, the modification of the material causes a refractive index increase, leading to light confinement and guiding. Alternatively, the demonstration of waveguiding by the double-line approach was provided in germanate (GeO_2_-PbO) and tellurite (TeO_2_-ZnO) glasses [106]. In this case, the fs-laser process leads to a stress-induced negative refractive index changes in the laser focal region, the light being guided in between the written lines.

Material structuring has been achieved not only with laser irradiation but also with ion beam irradiation. For example, a high-energy nitrogen ion beam was found to be also a suitable to fabricate waveguide. Berneschi et al. reported on the successful fabrication of such a channel waveguide in an active sodium-tungsten-tellurite glass [107]. The light confinement was achieved due to localized increase of the refractive index in the ion-implanted channel. The 2D light confinement was achieved due to localized increase of the refractive index in the ion-implanted channel. Nevertheless, due to their widespread accessibility, ease of operation and control, and high reproducibility, pulsed lasers are much more commonly used for fabrication of waveguide in glasses.

Beyond laser-induced structural modifications and associated density changes for integrated waveguides, as well as for NP precipitation for rare-earth emission enhancement, HMO can also undergo laser-induced dielectric crystallite phase transition. Indeed, the LaBGeO_5_ system is a model system, as it is one of the few oxides that easily forms glass as well as shows a congruent ferroelectric stillwellite crystal structure with large nonlinear optical properties [108]. Transparent ferroelectric crystallites can be obtained by thermal treatment for the LaBGeO_5_ system [108] or for other congruent systems langasite-type La_3_Ga_5_GeO_14_–Ba_3_Ga_2_Ge_4_O_14_ [109] Such crystallization can be obtained by continuous wave (cw) laser irradiation and local heating by resonant absorption of a doping rare-earth element such as Nd^3+^ ions in Nd_0.2_La_0.8_BGeO_5_ glasses with a cw 800 nm Ti:sapphire laser [110] or Sm^3+^ in Sm_0.5_La_0.5_BGeO_5_ Glass with a cw 1064 nm Nd:YAG laser [111]. It can even occur without any absorbing dopant by multi-photon absorption of high repetition rate femtosecond Ti:sapphire amplified lasers [112]. The optimal irradiation parameters and scanning speeds of these stoichiometric systems have allowed for the creation of single crystal precipitation [111], requiring the ideal management of thermal gradients at the voxel by the in-depth multi-plane management of spherical aberrations [113]. Moreover, directionally controlled 3D ferroelectric single crystals have been grown, showing a bended single crystal where the orientation of the ferroelectric c-axis follows the laser scanning direction [112]. Crystal-in-glass functionalities such as waveguiding and efficient nonlinear second harmonic generation have been demonstrated [111,114]. Three-dimensional crystal-in-glass architectures have thus been achieved, including integrated crystal Y-junctions compatible with a Mach–Zehnder geometry with losses no higher than 2.64 dB/cm at 1530 nm [112].

Moreover, it is demonstrated that the RE elements not only facilitate the laser-induced crystallization process, but also can enter the formed crystal lattice, which was investigated in detail for neodymium- and erbium-doped LaBGeO_5_ glass [110,115]. Finally, recent work brought a new understanding of the laser-induced crystallization process of LaBGeO_5_ glass and how to optimally manage heat deposition during laser irradiation in order to grow single-crystal waveguides in the glass of quasi-stoichiometric or even in non-congruent matrix compositions [116]. The results on laser-induced crystallization of LaBGeO_5_ glass demonstrate that this approach can be used for the implementation of crystal-in-glass photonic architectures and possess applicative perspectives in terms of fabrication of functional photonic circuits inside bulk glass materials or at the surface of ribbon-shaped fibers [117].

During laser irradiation, glasses can also undergo spinodal decomposition which leads to a three-dimensional interconnected texture. This interconnected texture of amorphous phases is produced by the heating and cooling cycle of the pulses. Spinodal decomposition was successfully obtained in tellurite glasses using an excimer laser (248 nm) [118]. These irradiated glasses could find various vital applications such as membranes and sensors—for example, if one of the phases can be selectively etched.

Radiation treatment can also be used to form bubble-like structures, which has been observed in several glass types, especially in silica glass [119]. As bubbles can be used to generate periodic structures, interest in controlled generation of these bubbles has grown recently [120]. Generation of bubbles in the glass with the composition of 85GeO_2_-15Na_2_O (mol.%) has been reported in [121]. The irradiation under a high repetition rate femtosecond laser creates a local melting and a migration of matter within the glass matrix. The spatial separation of Ge and O ions and the micro-explosion inside the glass melt are thought to be responsible for the generation of bubbles. Mobility of the bubble is followed by confocal Raman spectroscopy which highlights the thermal gravity convection and the viscous drag force effect during the process. Air-bubble-containing tellurite glass microspheres were successfully prepared by local heat treatment of tellurite glass cullets placed on a substrate using a cw-Ti:sapphire laser at the wavelength of 810 nm. This allowed for the selective formation of a bubble at a certain place in the microsphere [122]. Such micrometer-size spheres could find applications in micro-optical system for microlasers and microamplifiers, for example. Despite few studies on bubble generation in glasses, the origin of the bubble formation during radiation treatment is still unclear.

The combination of metallic nanoparticles with rare-earth ions in HMO-doped glasses, in order to enhance their luminescent properties as well as their nonlinear optical properties, has been of great interest during the past decades as discussed in the previous section. The formation of NPs during the radiation treatment was demonstrated for Bi-containing germanate glass using a 1030 nm, 370 fs, 500 kHz laser. Bi atoms were reported to migrate perpendicular to the laser irradiation direction inside the glass, and the observed Bi-enriched regions were associated with the formation of Bi nanoparticles inside the glass [103]. Despite the fact that various metallic particles can be precipitated in the glass, precipitation of Ag or Au NPs is highly desirable, as these are known to enhance the emission properties of REE ions in glasses. Local laser-induced precipitation of these particles was demonstrated and carefully studied for example in phosphate glasses [123] and few articles report similar behavior of laser-induced precipitation of Ag NPs or Au NPs in HMO glasses [124].

Recently, we initiated a study of silver-containing glasses in the TeO_2_-Na_2_O-ZnO-Ag_2_O system. We irradiated these glasses with an Ytterbium femtosecond oscillator (1030 nm central wavelength, 9.8 MHz repetition rate, 390 fs pulse direction (FWHM) with a 20x − NA = 0.75 microscope objective). We found that due to the glass matrix UV cutoff being located at larger wavelengths than the excitation bands of the embedded silver ions, silver chemistry did not seem to take place, contrarily to silver-containing phosphates. Nevertheless, glass modifications led to refractive index modifications compatible with waveguiding ability by means of the double-track approach. Indeed, a He-Ne beam at 632.8 nm was injected in such double-track modifications. The associated output beam profile is displayed in Figure 12a, revealing distinct waveguiding behaviors generated by two tracks separated by 10, 20, 30, and 40 µm, respectively.

Horizontal cross-sections of CCD images of the near-field output profiles for track separation of 10 and 20 µm and for track separation of 30 and 40 µm are provided in Figure 12b,c. The 10 µm separation shows a lossy mode with a non-confined profile outside of the two tracks. Larger separation distances of 30 and 40 µm show a poor spatial quality resulting both from the multi-mode waveguiding behavior and from the associated laser injection. For the ideal track separation of 20 µm, the induced structure behaves much more as a single-mode waveguide at wavelength of 632.8 nm, as generally demanded for integrated photonic applications. This result, in agreement with [14,107], confirms that, despite silver doping, the two-line “depressed cladding” configuration is favored in this system.

## 5. Conclusions and Future Opportunities

The studies on radiation-induced defects/effects in germanate and tellurite glasses were reviewed in this paper. Although a large number of studies on radiation of silicate and phosphate can be found in the literature, radiation treatment of tellurite and germanate glasses has not yet been deeply investigated. Radiation treatment of these glasses using different sources was found to lead to the formation of defects as well as structural modifications. The latter occurs due to three competitive processes: an effect of heat/temperature; effect of pressure; or induced local variation in chemical composition. However, the data on the origin of the structural changes induced by the radiation treatment are still fragmented and not well investigated. Nonetheless, this lack of the fundamental understanding has not hampered the practical use of the HMO glass structural modifications.

To summarize, the radiation treatment can lead to thermal and ionization effects. The thermal effects can include: (i) crystallization or spinodal decomposition which occurs due to localized heating; (ii) bubble formation due to the plasma formation which decomposes oxygen from glass matrix or due to the high temperature liberating dissolved gases; and (iii) ion migration. The ionization effects are (i) point defects due to the formation of free electron—hole pairs and (ii) generation of metallic NPs. With this review, we clearly show that radiation treatment is capable to modify macroscopic properties of HMO glasses, and these modifications can be controlled spatially, especially if made with use of high-intensity ultra-short pulsed laser radiation sources. At the same time, we would like to emphasize that the photoresponse of these materials should be investigated more, as a deeper understanding is required for fabricating materials with enhanced radiation resistance or enhanced sensitivity to radiation based on HMO glasses. Such knowledge would contribute to guiding the industries to manufacture new commercial mid-infrared transparent glasses with tailored photoresponse for use in the radiation environment.

## Figures and Tables

**Figure 1 materials-13-03846-f001:**
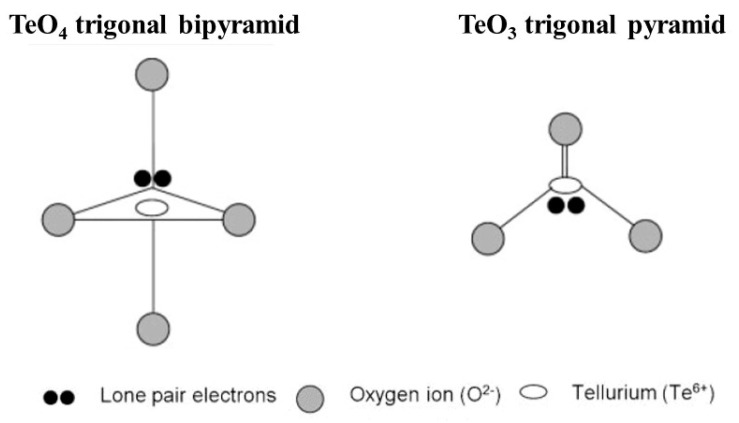
Schematic of the structural units in tellurite glasses, modified from [41].

**Figure 2 materials-13-03846-f002:**
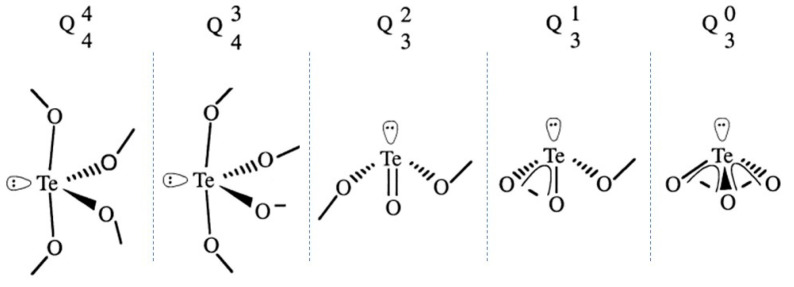
Schematic of the Q_m_^n^ units in the tellurite glasses, modified from [42].

**Figure 3 materials-13-03846-f003:**
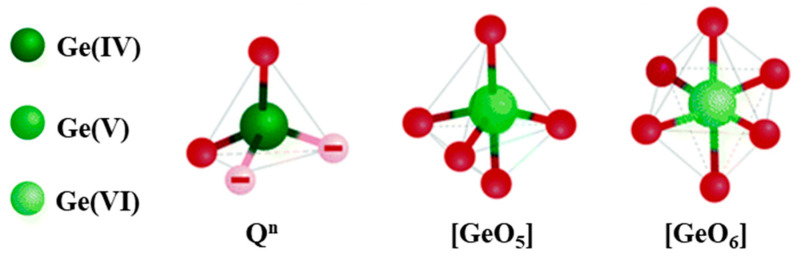
Schematic of the structural units in germanate glasses, modified from [46].

**Figure 4 materials-13-03846-f004:**
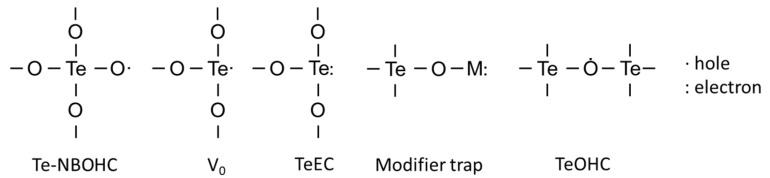
Defects in tellurite glasses.

**Figure 5 materials-13-03846-f005:**
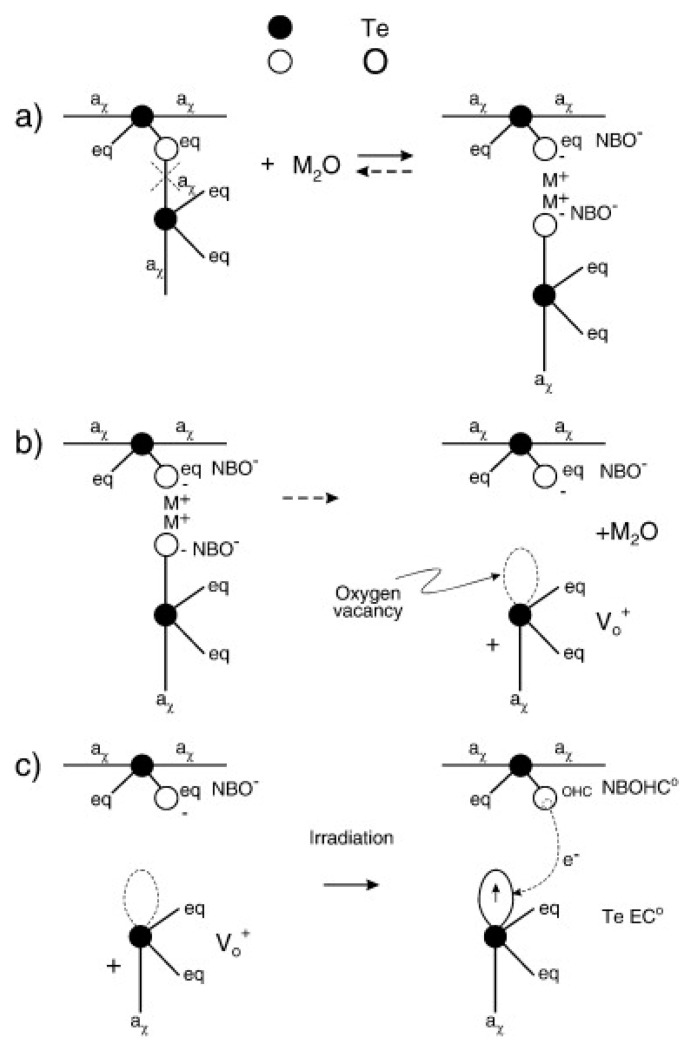
Formation of some defects. (**a**) Alkali oxide M_2_O and a BO react and form a pair of NBOs. (**b**) The reaction is partially reversed and a V_0_ defect forms. (**c**) Further irradiation results in electron transfer and formation of an Te-NBOHC and a TeEC defects. Reproduced from [62]. Copyrights Elsevier 2010.

**Figure 6 materials-13-03846-f006:**
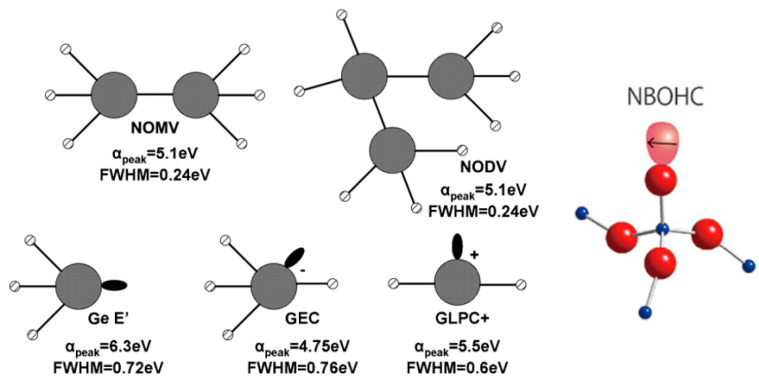
Various germanate defects with values of α and FWHM provided for each center correspond to position of the maximum and full width at half maximum of the absorption peak associated with the defect. Modified from [61,69].

**Figure 7 materials-13-03846-f007:**
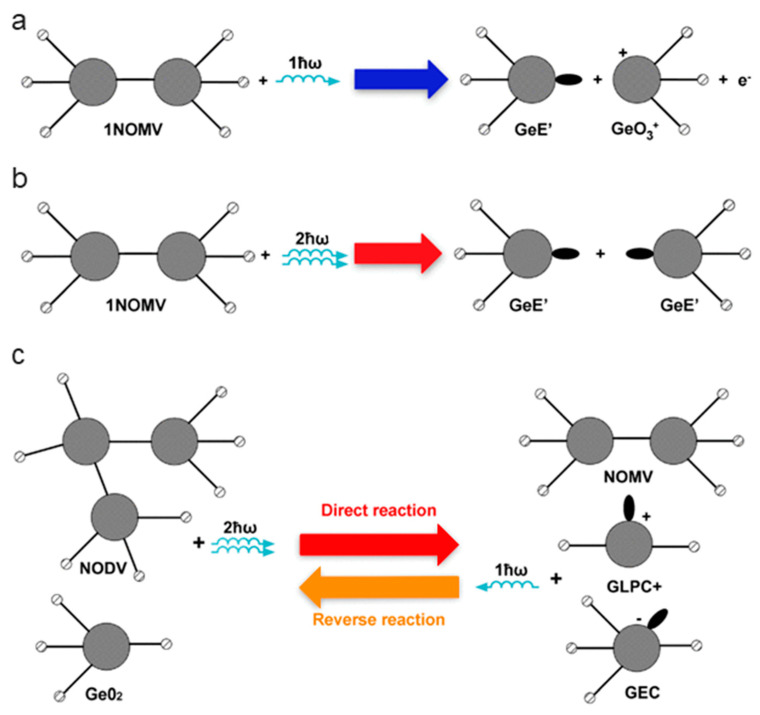
Proposed formation mechanisms of germanate defects. One-photon process (**a**), two-photon process (**b**), and balance reaction of one- and two-photon processes (**c**). Modified from [69].

**Figure 8 materials-13-03846-f008:**
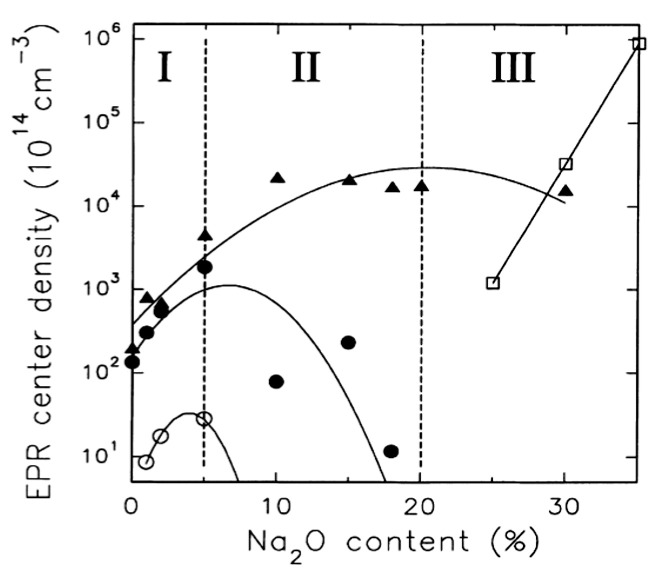
EPR center density in Na_2_O-GeO_2_ glasses as a function of the Na2O content. The hollow circles refer to Ge-E′ signals before X-ray irradiation, the filled circles refer to Ge-E′ signals after X-ray irradiation, the filled triangles refer to Ge-NBOHC signals after X-ray irradiation, and the hollow squares refer to the g = 2.21 signal (unidentified). The lines are drawn as a guide for the eye. Reproduced from [77]. Copyrights Elsevier 2000.

**Figure 9 materials-13-03846-f009:**
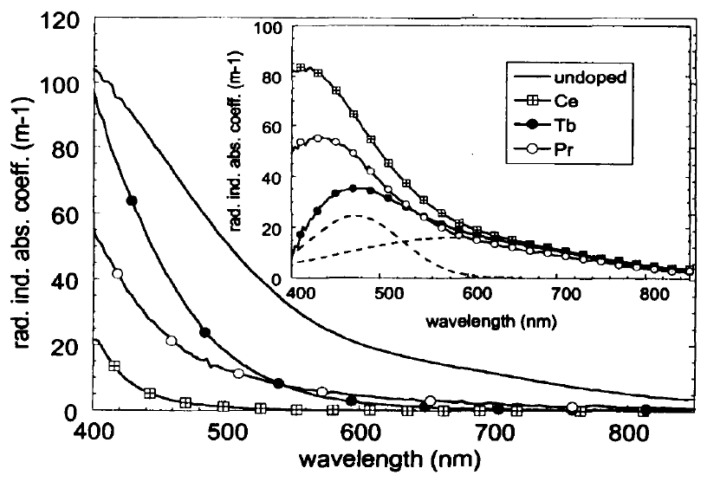
Radiation-induced color center distribution in all glasses after the highest irradiation dose. Inset: radiation-induced absorption bands “removed” by the trivalent doping and the Gaussian fit curves related to terbium doped glass (dashed lines). Reproduced from [80]. Copyrights Taylor & Francis 2003.

**Figure 10 materials-13-03846-f010:**
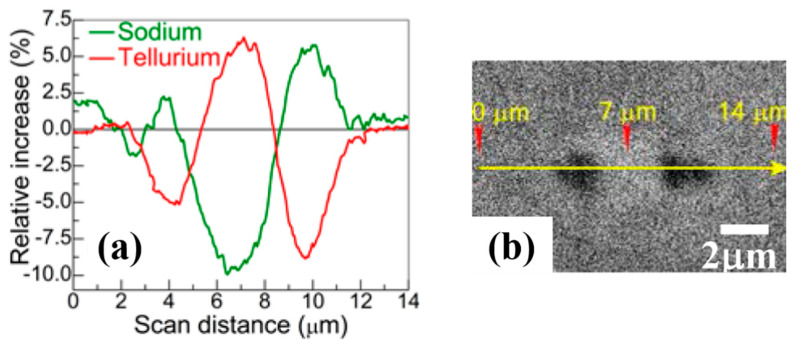
Line scans showing ion migration in 50TeO2-20P2O5-20Na2O-5ZnO-5ZnF2 (λ = 1040 nm, 400-fs, 39 nJ) (**a**) and the corresponding secondary electron image. (**b**) Modified from [94]: Copyrights OSA 2014.

**Figure 11 materials-13-03846-f011:**
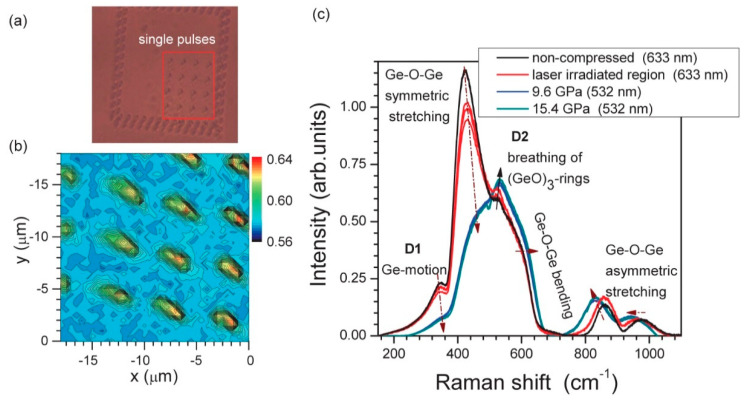
(**a**) An optical image of a GeO_2_ region modified by single pulses of 300 nJ/pulse energy (at the entrance of microscope), 800 nm wavelength, and 150 fs pulse duration focused at 10 μm depth. (**b**) Map of the region boxed in (**a**) at the 520 cm^−1^ D2-band, which corresponds to three-membered (GeO)_3_ rings. (**c**) Raman spectra of laser irradiated regions at different pulse energies 200, 300, and 400 nJ and at different hydrostatic pressures, measured using 532 and 633 nm wavelength illumination. Arrows in (**c**) shows the observed tendencies with increasing pulse energy and/or pressure. Wavelengths of laser irradiation for Raman measurements are denoted in the legend of (**c**). Reproduced from [97]. Copyrights OSA 2011.

**Figure 12 materials-13-03846-f012:**
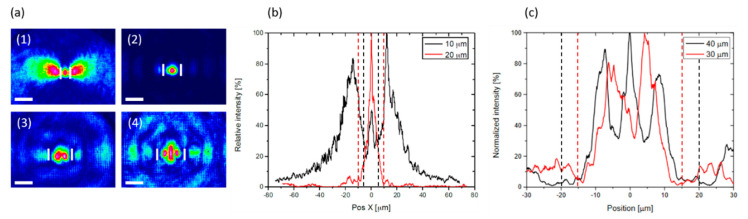
(**a**) Near-field helium-neon beam profile at the output of waveguides generated by two tracks separated by 10 µm (1), 20 µm (2), 30 µm (3), and 40 µm (4). Scale bar 20 µm, and white rectangles locate the double-track structure. (**b**,**c**) Horizontal cross-sections of CCD for track separations of 10 and 20 µm, and for track separations of 30 and 40 µm, respectively (unpublished results). Dashed lines indicate the respective positions of the double-track structures.

**Table 1 materials-13-03846-t001:** Defects found in tellurite glasses and their EPR values.

Defect	EPR g-Value
Te-NBOHC	1.9960 [62]
TeEC/V_O_	1.9705 [66]
Modifier related trap	2.0010 [63]
TeOHC	2.0747 [63]

**Table 2 materials-13-03846-t002:** Common defects found in germanate glasses, their EPR values, and absorption bands.

Defect	EPR g-Value	Abs. Wavelength nm [eV]	PL Wavelength nm [eV]
(GLPC)^+^	1.9866 [70]	225 [5.5] [69]	400 [3.1] [71] ^1^
GEC	1.9933 [70]	261 [4.75] [69] and 315 [3.9] [72]	-
Ge E′	2.0011 [73]	197 [6.3] [68]	590 [2.1] [61] ^2^
Ge-NBOHC	2.0076 [74]	375 [3.3] [72]	590 [2.1] [61] ^2^ 650 [1.9] [71] ^1^
GeO_3_^+^	2.008 [73]	-	-

^1^ Excited at 325 nm; ^2^ excited at various wavelengths between 250–400 nm.

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
