# Peer review of "Radiation-Induced Defects and Effects in Germanate and Tellurite Glasses"

_materials, 2020, doi:10.3390/ma13173846_

Round 1
Reviewer 1 Report
The work is a review paper focused on the on the radiation induced changes in germanate and tellurite glasses. In particular, the authors are interested in possible radiation-induced defects, mechanisms of their formation and the influence of the glass composition.
The paper is well written. It flows with a good sound and is properly designed.
I suggest only two minor considerations:
- corrections in some figures apparency
- 3, reduce the size of the test;
- 9, enlarge the figure because it is difficult to read
- fig 10, it could be useful to divide the figure into two parts, a and b, and enlarge the figure a because it is difficult to read;
- fig 11, enlarge the figures a and c because they are difficult to read;
- fig 12, use the partition (a, b and c), instead of I, ii and iii.
- due to germanate and tellurite glasses are two possible candidates in infrared field, it could be useful to discuss and to cite other works that investigate and apply different type of materials.
You can consider:
https://www.sciencedirect.com/science/article/abs/pii/S1005030210600850
https://www.osapublishing.org/ao/abstract.cfm?uri=ao-55-25-7142
Author Response
Reviewer 1
Reviewer:
I suggest only two minor considerations:
corrections in some figures apparency
fig 3, reduce the size of the test;
- Reorganized contents of Figure 3 while reducing text size and increasing clarity
fig 9, enlarge the figure because it is difficult to read
- Figure 9 is now enlarged
fig 10, it could be useful to divide the figure into two parts, a and b, and enlarge the figure a because it is difficult to read;
- Figure 10 is now modified: increased size, added a and b partitions. Text “Reproduced from …” in caption was changed to “Modified from …” and added references to “(a)” and “(b)” parts.
fig 11, enlarge the figures a and c because they are difficult to read;
- Figure 11 has higher resolution and is more readable.
fig 12, use the partition (a, b and c), instead of I, ii and iii.
due to germanate and tellurite glasses are two possible candidates in infrared field, it could be useful to discuss and to cite other works that investigate and apply different type of materials.
- Figure 12 was partitioned as suggested and Figure 12a was further repartitioned as (1), (2), (3), (4) instead of (a), (b), (c), (d). Included scale bars in 12a and dashed lines in 12b,c that were missing from previous manuscript.
Reviewer:
You can consider:
https://www.sciencedirect.com/science/article/abs/pii/S1005030210600850
https://www.osapublishing.org/ao/abstract.cfm?uri=ao-55-25-7142
Response:
Thank you for the link of these interesting papers. However, we do not think that these papers are relevant to our review:
- The first paper describes a study on germanate glass/glass-ceramic formation via standard thermal treatment method. While germanate glass crystallization is a part of this review paper, it is in the context of laser/radiation-induced crystallization, not bulk thermal treatment. Therefore, we believe that this paper would not provide sufficiently additional information within our context.
- The second paper describes thermal imaging of solid oxide fuel cells. The authors used thermal imaging in mid-IR, a spectral band suited for HMO glasses, in their measurements. However, no mention of glasses, high energy radiation or defects are made and thus does not contain useful information for our review.
Reviewer 2 Report
Mikko Hongisto et. al reported review article entitled on "Radiation-induced defects/effects in germanate and tellurite glasses" is interesting and well organised can be acceptable after minor revisions as noted below
1) Authors describes the defects/effects of laser radiations on germanate and tellurite glasses however authors need to keep some brief informations on laser radiations effects on doped/non-doped glasses with appropriate citations in the introduction part.
2) Better to tabulate the differences between the doped/non doped additives and their typical photo-physics (few characters emission/absorption ranges, two/one photon absorption/emission, chemical characteristics etc.) with respective glasses probably in the end. So that it will be very easy for the readers to grasp the concepts.
3) Conclusions and Perspectives can be more appropriate when authors explains the facts of reported works along with drawbacks. Also it is better to highlights the general laser induced changes in glasses to overcome drawbacks also for better photophysical performances.
4) Authors need to check for the high resolution images from online version of reported articles, some figures are not clear and hard to read as well. There should be many mistakes while keeping and organising figures, For example Fig. 9 is so unclear (two figures are merged hard to read inset figure, Fig. 10 has b) and b1) but nothing in captions there are many more. Please correct wherever its required.
5) Chemical naming and notations should be in IUPAC format and consistent. Please correct it wherever required.
6) Table 2, while providing g values please provide the formula in bottom of the table. Please provide excitation wavelength in PL spectral data (i.e for which wavelength of excitation they got λem maximum. If all are same, please provide in the bottom of table.
7) Please provide on plausible effects of radiations through schematic representation in Conclusion and Perspective sections.
Author Response
Reviewer 2
Reviewer:
1) Authors describes the defects/effects of laser radiations on germanate and tellurite glasses however authors need to keep some brief informations on laser radiations effects on doped/non-doped glasses with appropriate citations in the introduction part.
Response:
Added a sentence “With laser radiation treatment, the energy is controlled and focused and thus has been used extensively to write structures, such as active[13] and passive[14] waveguides in glasses, for example.”, starting on line 38.
2) Better to tabulate the differences between the doped/non doped additives and their typical photo-physics (few characters emission/absorption ranges, two/one photon absorption/emission, chemical characteristics etc.) with respective glasses probably in the end. So that it will be very easy for the readers to grasp the concepts.
Response:
Combined with comment 7, conclusion was edited to summarize the generation mechanism of each type of effects. We believe that there is insufficient data to provide a comprehensive table to for main glass/dopant combinations and therefore the focus kept in the different types of defects.
3) Conclusions and Perspectives can be more appropriate when authors explains the facts of reported works along with drawbacks. Also it is better to highlights the general laser induced changes in glasses to overcome drawbacks also for better photophysical performances.
Response: conclusion and perspectives section is now changed
4) Authors need to check for the high resolution images from online version of reported articles, some figures are not clear and hard to read as well. There should be many mistakes while keeping and organising figures, For example Fig. 9 is so unclear (two figures are merged hard to read inset figure, Fig. 10 has b) and b1) but nothing in captions there are many more. Please correct wherever its required.
Response: figures are now changed to high resolution and/or clearer ones.
5) Chemical naming and notations should be in IUPAC format and consistent. Please correct it wherever required.
Response:
After reviewing the chemical notations, 2 corrections in Figure 5 text were made. Here, 2 numbers were turned into subscripts. The commonly used notation in the glass science field was used elsewhere for readability for intended audience, although contrary to IUPAC (e.g. TeO3+1).
6) Table 2, while providing g values please provide the formula in bottom of the table. Please provide excitation wavelength in PL spectral data (i.e for which wavelength of excitation they got λem maximum. If all are same, please provide in the bottom of table.
Response:
Table is now modified to include this.
7) Please provide on plausible effects of radiations through schematic representation in Conclusion and Perspective sections.
Response: the conclusion and perspectives section is now modified. However, we decided not to use a schematic representation. We hope the reviewer will agree with our decision.
Reviewer 3 Report
- Overall the review paper is well written with adequate references
- Suggest the authors to introduce full text of abbreviations in the beginning of the manuscript, so it is easier for the reader
- for e.g. on page 2, line 54 MIR is not introduced.
- page 6, line 183 EPR.
- Once the full text is introduced, abbreviations can be used in the remainder of the text
Author Response
Reviewer:
Suggest the authors to introduce full text of abbreviations in the beginning of the manuscript, so it is easier for the reader
Response:
Thank you for the suggestion, however most of the abbreviations occurs in the same chapter as its full text and can be referenced quickly. Therefore, we believe that a full list is not needed for this paper.
for e.g. on page 2, line 54 MIR is not introduced.
Response:
On lines 55-56, MIR changed to mid-infrared as there are no more occurrences of MIR in the text.
page 6, line 183 EPR.
Response:
On line 187, the text now reads electron paramagnetic resonance (EPR)
Once the full text is introduced, abbreviations can be used in the remainder of the text
Response:
Requested changes made to the aforementioned parts. However, we expect the reader to know certain abbreviations, such as Nd:YAG and UV and thus are not introduced as it would make the text less readable.